# Assessing Consumer Preferences for Suboptimal Food: Application of a Choice Experiment in Citrus Fruit Retail

**DOI:** 10.3390/foods10010015

**Published:** 2020-12-23

**Authors:** Wen-Shin Huang, Hung-Yu Kuo, Shi-Yuan Tung, Han-Shen Chen

**Affiliations:** 1Department of Business Administration, Chaoyang University of Technology, No.168, Jifeng E. Rd., Taichung City 413310, Taiwan; wshuang@cyut.edu.tw; 2Department of Health Diet and Industry Management, Chung Shan Medical University, Taichung City 40201, Taiwan; jason87523@yahoo.com.tw (H.-Y.K.); phes10503@gmail.com (S.-Y.T.); 3Department of Medical Management, Chung Shan Medical University Hospital, No. 110, Sec. 1, Jianguo N. Rd., Taichung City 40201, Taiwan

**Keywords:** food choice motivations, food waste, willingness to pay, consumer behavior

## Abstract

Amid the trend of sustainable development, reducing food waste is a global concern and campaigns to reduce food waste have been launched. For example, the term “food sharing” has originated from Germany and promotes sharing food instead of wasting. “The Guerilla Kitchen”, which originated from Netherlands, is an organization that also promotes avoiding wasting food. Consequently, more and more people are paying attention on this issue and we think it is necessary to understand people’s acceptance of suboptimal food, as discarded suboptimal food represents a significant proportion of food waste. Additionally, at least one-third of the food globally produced each year is classified as suboptimal and cannot be sold in the market because of a poor appearance, damaged packaging, or near expiration date, thus presenting challenges for environmental, social, and economic sustainability. Previous studies on suboptimal food have focused more on appearances and packaging dates and less on investigating traceable agricultural and price discounts, which is where food classified as suboptimal entails a discount. Moreover, citrus product attributes such as appearance, size, freshness indicators, traceable agricultural products, and price discounts were determined in terms of consumer preference through pre-measurement here, then using a choice experiment method to clarify which attributes consumers care about most (*N* = 485 respondents). Conditional logistic regression and a random parameter logit model (RPL) are employed to examine the various properties of a marginal willingness to pay (WTP). RPL was also used to deduce the respondents’ choices based on differences in appearance and freshness indicator. The results showed that consumers place greater emphasis on the freshness indicators (harvesting/packaging date labels) and appearance of suboptimal citrus fruits but do not focus on the size. Consumers are willing to purchase citrus fruit with a flawed appearance, although the price needs to be reduced from the original price. Although suboptimal food does not reduce health, people may still not buy it and this result in food wastage. As a result, it is essential to increase awareness regarding suboptimal foods and reduce food waste to support sustainable development.

## 1. Introduction

According to United Nations statistics, nearly one-third of the total international output of food is wasted every year, and the annual cost of food waste disposal is as high as 940 billion US dollars [1]. Kretschmer et al. (2013) [2] highlighted that according to the data of the US Food and Agriculture Organization in 2013, about three-quarters of food is wasted at production sites, households, and restaurants, and household food waste has increased over time. About 15.9% of food waste in the USA comes from the consumer, and USA households produced 27 million tons of food waste in 2015 [3]. Additionally, on the basis of food waste, the Environmental Protection Administration of the Executive Yuan (2018) [4] estimated that about 3.68 million tons of food materials are consumed in Taiwan every year, and an average of 158 kg is wasted per person, of which nearly 50% is discarded by individuals and families. These numbers were higher than those of other Asia-Pacific countries, thereby indicating that Taiwanese people waste too much food [4]. Stuart (2009) [5] and Bilska et al. (2016) [6] suggested that the proportion of waste at the supply chain is often higher than that at the consumer side and that the sources of food waste in the supply chain include mislabeling outer packaging, poor product appearances, being near the expiration date, and outer packaging damage. Lebersorger and Schneider (2014) [7] pointed out that a significant proportion of food is wasted at the retail stage that is disposed while in good condition, and said food is only discarded based on an expiration date that has been passed (e.g., in Austria, more than a quarter of discarded food and products are suboptimal products). Lombart and Louis (2014) [8] argued that selling suboptimal products can lead to a positive effect on the image of the store as a being responsible stakeholder, with the potential to influence consumer’s retail preferences or loyalty. Plazzotta et al. (2017) [9] pointed out that fruit and vegetable waste is mainly generated before reaching consumers due to programmed overproduction and the unfulfillment of retailer quality standards. Therefore, retailers or companies may not purchase products that cannot reach their standards. Aschemann-Witzel et al. (2017) [10] stated that more than one million people in the world experience chronic hunger every year and some regions even face food crises, resulting in a serious global food imbalance. Aschemann-Witzel et al. (2015) [11] stated that food waste has an impact on environmental, social, and economic sustainability. Thus, food waste has been regarded as a moral issue due to global inequality in food access and growing food security issues [11,12,13], and it is indeed necessary to solve food waste challenges to achieve sustainable development in food supply chains [14].

The Homemaker’s United Foundation (2016) [15] highlighted that suboptimal food, which is unmarketable because it does not meet traditional aesthetic standards, mainly includes vegetables, fruits, and meat. These items usually do not procure good sales due to their poor appearance and can only be used as feed, fertilizers, canned juices, donations for food banks, or even thrown away. Suboptimal food is divided into three categories on the basis of its characteristics, namely, appearance standards (e.g., weight, shape, and size are required to meet the ideal standards) [16], the marked expiration date (e.g., food approaching or exceeding its expiration date) [17], and the packaging (e.g., food packaging exhibiting visual damage, such as a dented can or a torn wrapper) [17]. Additionally, it is necessary to confirm that these categories/aspects do not pose any safety risks and that the food is still appropriate for human consumption [11,18,19].

Dion et al. (1972) [20] stated that when selecting foods, most consumers select foods with a perfect appearance and shape, undamaged packaging, and a long shelf life, thus resulting in suboptimal food that cannot be sold. Göbel et al. (2015) [18] suggested that the main reason for the waste of vegetables and fruits is the influence of retailers on product quality standards and specifications. White et al. (2016) [17] revealed that consumers may choose less suboptimal food because of factors such as an abnormal shape, damaged packaging, or marked date of expiration. Helmert et al. (2017) [21] also revealed that only a few consumers will choose suboptimal food when the quality or safety of suboptimal food is similar to that of optimal food. Symmank et al. (2018) [22], with the help of an example, stated that the appearance of bananas will affect German consumer purchase intentions and that they attach importance to the shelf life of bananas. Aschemann-Witzel et al. (2018) [23] highlighted that printing requests/instructions such as “no food waste” on food packaging can increase the possibility of Uruguayan consumers choosing suboptimal food. Halloran et al. (2014) [19] highlighted that food waste could be reduced through communication and improved food packaging and labeling.

Additionally, consumers will consider the price when choosing products. Helmert et al. (2017) [18] pointed out in their research that price badges can influence the attention of European consumers, cognitive processing, and purchase intentions regarding suboptimal food. Many retailers in Europe offer products close to the shelf life at lower prices to attract consumers [11]. According to Grewal et al. (1998) [24], retailer price discounts can affect consumer purchase intentions. Verghese et al. (2013) [25] also pointed out that the precondition for consumers to buy suboptimal food is a price discount. The study of de Hooge et al. (2017) [26] stated that consumers in Denmark, Germany, Norway, Sweden, and the Netherlands are willing to buy suboptimal food with price discounts. Aschemann-Witzel (2018) [27] pointed out that when the price of suboptimal food is reduced, consumers may have more incentives to buy them. De Pelsmacker et al. (2005) [28] believed that when discussing consumer purchase behaviors, they cannot be judged solely by their attitude and preference toward the product but must be analyzed in terms of their purchase intention or willingness to pay (WTP). The so-called WTP is the amount that consumers are willing to pay for a product that they think is most appropriate [29]. Tsiros and Heilman (2005) [30] revealed that consumer WTP for a product will decrease with the shortening of its shelf life. The research results of Nandi et al. (2016) [31] also pointed out that Indian consumers are willing to pay a higher WTP for fruits and vegetables grown in an eco-friendly way. Previous studies on consumer WTP for suboptimal food show that price discounts will attract purchase decisions [10,25,32]. Since there is presently no definite range for price discounts for suboptimal food in Taiwan, this study includes price discounts as an attribute variable to explore the prices that Taiwanese consumers are willing to pay for suboptimal food.

The contingent valuation method (CVM) is often used to evaluate consumer WTP for non-market goods. The CVM asks respondents about their WTP for a certain good in a hypothetical market through a questionnaire survey [33,34]; however, because the CVM has some limitations in terms of application, a strategic bias may be caused because of overestimation or underestimation. For example, respondents deliberately conceal their real preference for non-market goods for their own interests [35]. Therefore, the choice experiment method (CEM) has gradually become an important evaluation tool for measuring the value of non-market goods [36]. The CEM was first proposed by Louviere and Hensher (1982) [37] and Louviere and Woodworth (1983) [38]. Its theoretical framework is derived from random utility theory [39]. CEM has been widely used in non-market value evaluation in recent years. For example, Tait et al. (2016) [40] used a CEM approach to explore the WTP for mutton products with an environmental label certification for consumers in Britain, China, and India. Scarborough et al. (2015) [41] used a CEM to discuss British consumer preferences for products with traffic light labels in supermarkets. Ortega et al. (2015) [42] used a CEM to study Chinese retailer preferences for food quality and safety attributes. Grebitus et al. (2015) [43] used a CEM to explore the roles of human values and generalized trust on stated preferences when food was labeled with environmental footprints. Meyerding et al. (2019) [44] used a CEM to explore German consumer preferences for product attributes (e.g., place of origin and production method) and WTP for local fresh tomatoes and their processed products (e.g., ketchup). Thøgersen et al. (2019) [45] used a CEM to explore German, French, Danish, Chinese, and Thai customer preferences for product attributes (e.g., the country of origin, organic badge, and price) for organic foods produced in the corresponding countries.

In summary, previous studies on suboptimal food have mainly focused on the appearance, date, and packaging, as well as consumer preferences for different types of suboptimal food. Taiwan is known by many as the “kingdom of fruit.” According to the Yearly Report of Taiwan’s Agriculture 2018 Supply and Demand for Food by the Council of Agriculture of the Executive Yuan, citrus represents the largest fruit production category in Taiwan (524,087 metric tons). Therefore, citrus is used as the main product for investigation in this study. In addition, the appearance of citrus fruit is limited by fungal decay [46] and peel defects [47] during store retailing, affecting the willingness to pay. The contribution of this study lies in dividing the attributes of suboptimal food in terms of the appearance, size, freshness indicator, traceable agricultural products, and price discount and using a CEM to deduce the overall preferences of consumers and the consumer WTP for various product attributes for suboptimal food. This is achieved via conditional logit (CL) and random parameter logit model (RPL) analyses. It is expected that the research results can be used as a reference for retailers as well as other sales channels and enhance the public’s awareness of suboptimal food. When consumers purchase citrus, the appearance, size, freshness indicator, and price discount are the primary factors which impact the purchase intention. On the other hand, a traceable agricultural seal is generally used for agricultural products in Taiwan, which is why this attribute is discussed in terms of suboptimal fruit here.

## 2. Materials and Methods

### 2.1. Survey Design

This study explores the product attributes of suboptimal citrus fruit, including the appearance (complete and defective), size (large, medium, and small), freshness indicator (labeled and unlabeled), traceable agricultural products (with and without certification), and price discount. The variety of the citrus fruit is *Citrus poonensis,* which is the most common in Taiwan. The details are shown in Table 1.

To develop an easy method for respondents to fill out questionnaires, this study adopted an orthogonal design which was submitted through Statistical Product and Service Solutions. Ninety-six (23 × 31 × 41) sets were selected, and after factoring out the redundancies, each choice set contained two random number substitutes and one status quo which included flawless appearance, medium size, no freshness indicator, uncertified traceability, and a discount price of 40 New Taiwan (NT) dollars. Each survey included three choice sets extracted from among them, with a total of 15 possible questionnaire versions.

In this study, we used judgmental sampling to survey questionnaire answers face-to-face by paper and pencil in supermarkets, mass merchandisers and traditional market. First, the study conducted a pre-test questionnaire, with the aim of understanding consumers’ overall consumption preferences and WTP for suboptimal food. The questionnaires were issued from 1 March 2020 to 31 March 2020 to consumers who had purchased suboptimal food in the past six months. During the first stage, 150 questionnaires were issued, out of which 121 were valid, and the effective questionnaire recovery rate was 80.67%. The official survey was divided into three parts. Part one included the degree of importance that consumers place on the suboptimality of five cases of citrus fruit, asking consumers to rank the issue in importance from 1 to 5 according to their own personal beliefs. Part two presumes that the consumers are going to purchase citrus products with various attributes (i.e., overall superior fruit) and provides plans for citrus products based on their appearance, size, freshness indicator, and food traceability seal, then assigning a discount to be displayed to help consumers choose the one they like best based on their own personal preferences toward the attributes of suboptimal food (as shown in Figure 1).

Part three is the demographic information of the survey respondents, which includes their gender, age, education, marital status, monthly income, and whether they would consider buying products with a traceability seal or scanning QR codes that can allow pertinent information about the product to be read.

### 2.2. Choice Experiment Method

A CEM approach is used to establish a hypothetical market to investigate consumer preferences for non-market goods. As the CEM has the ability to evaluate multiple attributes and levels, different alternatives are combined for the important characteristics related to non-market goods or services. Through the choice sets with different situational assumptions, respondents could select appropriate alternatives according to their preferences to avoid errors in the evaluation. In regard to the empirical model, the conditional logit (CL) regression model is used to estimate consumer average preferences for multiple attributes for fresh food, as well as their marginal willingness to pay (MWTP) for the attribute levels [48]. Secondly, the random parameter logit model (RPL) can be used to explore the preference heterogeneity and the WTP for different characteristic attributes for respondents with different socioeconomic backgrounds [49]. Bazzani et al. (2017) [50] used the CEM to explore Italian consumer preferences for various product attributes (e.g., product sources and production methods) and WTP for local and organic foods. Kallas et al. (2019) [51] discussed Spanish consumer purchase intentions and WTP for innovative patties containing black pork products enriched with porcini mushrooms as a natural source of dietary fiber or blueberries as a natural antioxidant source. Ceschi et al. (2017) [52] investigated Italian consumer preferences for product attributes for apples, for example, being organic, their color (bicolored, green, and red), origin, and import country, also evaluating their WTP.

First, this study adopted the CEM to construct a preference and utility model for suboptimal citrus fruit, then applying the CL and RPL to estimate the utility function for preferences, then finally exploring the MWTP attributes in terms of the demographic information of the respondents. In the binomial model below, “j” is the utility function that was arrived at for the hypothetical respondents “i” through a product substitution, as in Formula (1).
(1)$$U_{\mathrm{ij}}=V_{\mathrm{ij}}+\varepsilon_{\mathrm{ij}}$$
where $U_{\mathrm{ij}}$ represents the attribute of the *i*-th respondent facing the *j*-th option, $V_{\mathrm{ij}}$ represents the observable part of the utility function, and $\varepsilon_{\mathrm{ij}}$ represents the residual item, i.e., the unobservable random utility.

The hypothesis assumes $V_{\mathrm{ij}}$ to be the linear form of the substitute’s observable attributes $X_{\mathrm{ij}}$; thus, it is possible to take the consumer utility i contained in the k item attributes of substitute j and assume the price variable of that set of attributes to be $P_{j}$, therefore the consumer utility function i can be expressed as Formula (2):(2)$$U_{\mathrm{ij}}=V_{\mathrm{ij}}+\varepsilon_{\mathrm{ij}}=\overset{k}{\underset{k=1}{\sum }}\alpha_{k}X_{\mathrm{jk}}+{\beta P}_{j}+\varepsilon_{\mathrm{ij}}$$
where $U_{\mathrm{ij}}$ is the utility that consumer i derives from product j; $X_{\mathrm{jk}}$ is the *k*-th attribute of the product j, and $P_{j}$ is the price of product j. In additional, $\alpha_{k}$ and β are the parameters to be estimated.

In order to explore the reasons behind respondent preferences for certain parameters for suboptimal citrus fruit, Formula (2) was expanded into Formula (3), which adds in the random utility function of the respondent demographic information:(3)$$U_{\mathrm{ij}}=\overset{k}{\underset{k=1}{\sum }}\alpha_{k}X_{\mathrm{jk}}+\sum_{k}^{K}\sum_{q}^{Q}\gamma_{\mathrm{kq}}X_{\mathrm{jk}}Z_{\mathrm{iq}}{\beta P}_{j}+\varepsilon_{\mathrm{ij}}$$

In the formula, $U_{\mathrm{ij}}$ is the utility that the *i*-th respondent derives from the product j and $X_{\mathrm{jk}}$ is the k-th attribute of the product j. $Z_{\mathrm{iq}}$ is the qth demographic information of the respondent i. $\alpha_{k}$ is the attribute variable coefficient. $\gamma_{\mathrm{kq}}$ is the overlap coefficient of the attribute variable and demographic information.

By separating the price variables in Formula (3), it can be seen clearly and the price that consumers are willing to pay may be more easily analyzed by Formula (4):(4)$$U_{\mathrm{ij}}=V_{\mathrm{ij}}\left(X_{\mathrm{ij}},S_{i}\right)+\varepsilon_{\mathrm{ij}}$$
where $V_{\mathrm{ij}}$ is the utility coefficient of observable variable $X_{\mathrm{ij}}$ and respondent characteristic $S_{i}$, which represents the respondent’s preference, and $\varepsilon_{\mathrm{ij}}$ is the residual item.

To measure the WTP for the product attributes, we took the total differential of Formula (2), treated the utility as a constant, and assumed that ${\mathrm{dU}}_{\mathrm{ij}}=0$, which gave Formula (5):(5)$${\mathrm{dU}}_{\mathrm{ij}}=\sum_{k=1}^{K}\alpha_{k}{\mathrm{dX}}_{\mathrm{jk}}+{\beta \mathrm{dP}}_{j}=0$$

When other attribute variables remain constant (${\mathrm{dX}}_{j1}={\mathrm{dX}}_{j2}=\cdots ={\mathrm{dX}}_{\mathrm{jk}-1}=0)$, finding the consumer WTP for $X_{\mathrm{jk}}$ attribute(s) of product j can be carried out as per Formula (6):(6)$${\mathrm{dU}}_{\mathrm{ij}}=\sum_{k=1}^{K}\alpha_{k}{\mathrm{dX}}_{\mathrm{jk}}+{\beta \mathrm{dP}}_{j}=0$$

## 3. Results and Discussion

### 3.1. Sample Size and Composition

To understand the attributes of the respondent consumption preferences for suboptimal citrus fruit, this study utilized judgmental sampling to target consumers who have purchased suboptimal food. A total of 670 questionnaires were issued. After factoring out invalid questionnaires, a total of 485 valid questionnaires were obtained, representing a 72.4% questionnaire recovery rate. The screening rule for invalid questionnaires was when respondents did not ever purchase suboptimal food. In such cases, the questionnaires were classified as invalid questionnaires. The largest respondent proportion was males (51.1%). Age was primarily concentrated in the 41–50 age range (27.8%), followed by 51–60 years of age (25.8%), and 31–40 years of age (22.7%), showing that the middle-aged demographic is more likely to buy suboptimal food than other consumer groups. In terms of education level, the proportion with a tertiary education was the highest (48.5%). In terms of marital status, the majority of respondents were married (57.5%). The average monthly incomes of individuals were primarily in the NT$ 40,001–60,000 range (35.1%), followed by the NT$ 20,001–40,000 range (28.0%). More than half (57.7%) of consumers had purchased traceability-certified products, but most did not scan the QR code to read the pertinent information. The respondent purchasing locations for suboptimal food (detailed below) were mainly supermarkets (73.8%), followed by mass merchandisers (47.4%), which are companies that affordably sell large quantities of goods that appeal to a wide variety of consumers, and traditional markets (42.1%). The most purchased foods (detailed below) were vegetables (57.7%), followed by fruits (53.0%), and whole grains and tubers (51.1%).

### 3.2. Emphasis on Attributes

The results of the research show that the degree of emphasis placed on the various attributes was highest for freshness indicators (3.86 points), followed by appearance (3.22 points), traceability certification (2.66 points), price discounts (2.64 points), and size (2.62 points). The results of the weight comparison analysis show that consumers place greater emphasis on the freshness indicators and appearance of suboptimal citrus fruits but do not focus on the size, which is different from the results of the study by de Hooge et al. (2017) [26], which found that consumers in Germany, the Netherlands, and Norway place great emphasis on price discounts for suboptimal food. It is presumed that the aforementioned countries rely on imports due to climatic and environmental factors that affect the types and quantities of fruits and vegetables that may be locally grown, therefore emphasizing pricing changes.

### 3.3. Preferred Suboptimal Products by Consumers

This study analyzed the 11 most preferred attribute sets included in the suboptimal citrus fruits as selected by the respondents. The results show that the most preferred set of attributes includes a flawed appearance and moderate size with a freshness indicator, traceability certification, and a discounted price of NT$ 25 (accounting for 11.48% of respondents). The second most common set was the set with a flawless appearance, large size, freshness indicator, no traceability certification, and a discounted price of NT$ 25 (accounting for 10.65%). The least preferred attribute set by consumers was a flawed appearance with a discounted price of NT$ 25 (2.27%) and a flawed appearance with a discounted price of NT$ 35 (0.96%). It is presumed that a possible reason for this is that consumers themselves are not willing to buy citrus fruits with a flawed appearance while paying little attention to discounts and will only buy them in the hopes that, flawed appearance notwithstanding, there are other certifications that can guarantee the product.

### 3.4. Conditional Logit Analysis Results

Based on random utility function given by Formula (1), a utility model for suboptimal citrus fruits was established to understand consumer preferences for suboptimal citrus fruits, as in Formula (7):(7)$$U_{\mathrm{ij}}=ɑ1\mathrm{EDij}+ɑ2\mathrm{SZ}1\mathrm{ij}+ɑ3\mathrm{SZ}2\mathrm{ij}+ɑ4\mathrm{FRij}+ɑ5\mathrm{TAPij}+\beta \;\mathrm{FUNDij}+\varepsilon \mathrm{ij}$$

In the formula, i = 1, 2, 3, …, 485, which is the total sampling of 485, and j = 1, 2, 3, …, 12, which represents the 12 choice sets for the suboptimal fruit attributes.

A coefficient for the attribute variable was estimated for Formula (7) through NLOGIC 4.0 Conditional Logit, and then the coefficient value was substituted into Formula (7) to find the discounted prices that were willing to be paid for each attribute. The empirical estimation results are summarized in Table 2.

In terms of the levels for suboptimal citrus fruit attributes, appearance, freshness indicators, and traceability certifications were all at the 1% significance level, while only a small size (SZ2) was at a 10% significance level, meaning that willingness to consume is affected by whether the appearance is flawed, size is too small, and whether the product has freshness indicators and traceability certifications. Maintaining the status quo (ASC) was both positive and significant at a 5% significance level, indicating that consumers would prefer to maintain the status quo.

Second, a coefficient was estimated through the utility function from Formula (1) and was substituted into the theoretical model (Formula (6)) to calculate the WTP. The prices that were willing to be paid for the attributes were in the following order: appearance (NT$18), small size (NT$36), freshness indicator (NT$68), and traceability certification (NT$63). The results of the analysis revealed that, of the four suboptimal citrus attribute levels, the price discount for appearance is the highest, which means if sellers want consumers to buy suboptimal food, the price needs to be reduced from the original NT$40 to NT$18. Additionally, based on the results of the study, consumers are willing to pay more for citrus fruits with freshness indicators and traceability certifications, which indicates that consumers prefer these two product attributes. Furthermore, because consumers care about the freshness indicators the most, no matter how much the discounted price is, they will not purchase fruit without a freshness indicator. This is because there are no other guarantees such as freshness indicators or traceability certifications on the suboptimal products, and consumers may believe that these fruits could endanger their health. According to a study from de Hooge et al. (2018) [53], fruits, vegetables, and foods with dented packaging should not be regarded as inferior products. Although they are visibly different from the best products, visual flaws are considered a sign of authenticity. Tsalis (2020) [54] believes that retailers in most countries and regions only sell suboptimal food as a cheap product that does not legitimately generate purchase intentions. Wang et al. (2018) [55] pointed out that product certification labels can eliminate the uncertainty that consumers face when buying products. Thus, WTP for products with certification labels will increase accordingly.

### 3.5. Random Parameter Logit Analysis Results

Since the CL assumes that the parameters in the respondents are fixed, the average preferences of the respondents were evaluated, while the RPL is based on the attribute parameters of the respondents taking the form of a normal distribution, where the differences in preferences for the suboptimal citrus attributes can be evaluated. The results of using the CL and RPL to evaluate suboptimal citrus attributes were quite dissimilar. The RPL presented respondent preferences for appearance, larger or smaller sizes, freshness indicators, and traceability certifications, while the CL presented respondent preferences for appearance, small size, freshness indicators, and traceability certifications, except for large sizes. Additionally, the RPL also reflects the heterogeneous distribution of respondent preferences for various attribute parameters. As shown in Table 3, keeping the status quo was found at a significance level of 1%, and appearance and freshness indicators are both significant and indicate heterogeneity in respondent preferences between appearance and freshness indicators. This means that consumers cared more about appearance and freshness indicators for suboptimal products than other attributes.

The coefficient value estimated through the utility function (Formula (1)) was substituted into the theoretical model (Formula (6)) to calculate the respondent WTP. The prices for each attribute were in the following ascending order: appearance (NT$19), larger size (NT$36), smaller size (NT$33), freshness indicators (NT$64), and traceability certifications (NT$59). The analysis results show that, of the five attribute levels for suboptimal citrus fruit, a price discount for appearance ranks the highest, which means that consumers are willing to purchase citrus fruit with a flawed appearance, although the price needs to be reduced from the original price of NT$40 to NT$19. The freshness indicator and traceability certification results show that consumers are willing to pay more for citrus fruit with these two product attributes. Freshness indicators represent the highest price increase, changing from the original price of NT$40 to NT$64. Jaeger et al. (2018) [56] pointed out that the product appearance, aroma, expiration date, and overall sensory evaluation are the purchase intention determinants. Hingston and Noseworthy (2020) [57] pointed out that consumer aversion to agricultural products with an abnormal appearance depends on their personal experience with these foods. The factors which impact consumer purchase intentions for food are inferences about taste, texture, and safety. Van Boxstael et al. (2014) [58] pointed out that most consumers have different opinions on shelf life labels and expiration dates for different food types.

### 3.6. Exploration of Respondent Demographic Information in Suboptimal Food WTP Heterogeneity

The results of the RPL analysis show that there were random parameters for the appearance and freshness indicators. Therefore, this study compared WTP with the respondent demographic information based on the two aforementioned attributes. The results of this analysis are shown in Table 4.

There was a significant difference in terms of the average monthly income of individuals who were willing to pay for appearance. Of them, respondents with an average monthly income between NT$40,001 and NT$60,000 were willing to pay a lower price, indicating that those with an average monthly income of the middle class are less willing to buy products with a flawed appearance. Of these respondents, those aged between 21–30 years with a tertiary education or above and those whose monthly income was between NT$20,001 and NT$40,000 were willing to pay a higher price. This shows that young people and those with higher education and a monthly income from NT$20,001 to NT$40,000 more greatly emphasize product freshness and are therefore willing to pay more for products that display freshness indicators. This result is consistent with the study by Tsakiridou et al. (2011) [59] that identified the consumers who are willing to pay higher prices for fruits with food safety labels.

## 4. Conclusions

### 4.1. Concluding Remarks

The study results show that, of the suboptimal citrus fruit certification attributes, the most important is the freshness indicator, followed by appearance, traceability certifications, price discounts, and finally size. Based on the results found here, the suboptimal citrus attribute set most preferred by respondents was the following: appearance flaws, moderate size, freshness indicators, traceability certifications, and a discounted price of NT$25. The second-most preferred attribute set was the following: perfect appearance, larger size, freshness indicators, without traceability certifications, and a discounted price of NT$25. The least preferred certification plan was the one with only appearance flaws and a discounted price of NT$35, along with the one with only appearance flaws and a discounted price of NT$25. This was presumably due to consumers not being willing to buy suboptimal citrus fruits with a flawed appearance and paying less attention to price discounts, although, a flawed appearance notwithstanding, there are other certifications that can provide a quality guarantee for the products. Thus, retailers should upgrade their food preservation systems to keep products fresh. In terms of primary producers, they can used suboptimal foods in food processing such as the production of fruit jams, canned vegetables, candied fruits, etc.

### 4.2. Recommendations

#### 4.2.1. Managerial Implications

This study has analyzed the importance of each fruit attribute based on respondent preferences and has found that consumers are not overly focused on size. The reason for this is presumed to be due to the fact that citrus products purchased by consumers in the market are classified through a screening mechanism before being circulated and sold in the market. The screened products are mainly medium-sized and above (25−30 cm), with smaller sizes being rejected and removed before reaching consumers. Therefore, it is recommended that relevant government agencies provide publicity and explanations for promoting suboptimal food being processed to change its form, such as making it into juice or canned food, thus greatly increasing its value. Governments should relax any regulations on the minimum sizes of fruits as consumers will still buy smaller fruits. Mass media promotion can promote consumers to buy and eat fruit, which can not only reduce losses for growers, but also reduce food waste. Besides emphasizing freshness indicators, appearance is another important attribute. The reason for this is presumably due to the inability of consumers to accurately determine the quality of fruit. Therefore, the appearance attribute is the second priority. If a product has a freshness indicator, this represents a guarantee for both the product and the consumer.

The results of the empirical analysis here show that consumers who prefer suboptimal citrus fruit with a freshness indicator and traceability certification are willing to pay more for the purchase. Therefore, it is recommended that the government not only stipulates that packaged foods need to show an expiration date, but also advocates for the popularization of freshness indicators for bulk foods or the addition of packaging. For example, using the Kanban software to indicate harvest dates and making the label certification process more transparent, i.e., label certifications and date indications can be displayed on product packaging. Meanwhile, more food-related knowledge should be spread to enhance public awareness. Under the assumption that a product is guaranteed, consumers can make discerning purchases that are not just for gifting or personal use based on their product recognition. Products with appearance flaws have the highest price discounts. Currently, there is no clear range of discounts in the Taiwanese market. Fresh ingredients in supermarkets and hypermarkets on site will be sold at discounts (20−40% off) based on their expiration dates. It is recommended to plan to establish a discount system based on the characteristics of the food category or the expiration date and provide references for retailers or other sales channels.

#### 4.2.2. Research Limitations and Future Research Direction

There were a number of limitations in the research process here. If it is possible to expand the scope of future research, then the research framework may be perfected. This study makes the following recommendations in connection with the conclusions and limitations of this research. Only five attributes for suboptimal food (appearance, size, freshness indicator, traceability certificates, and price discounts) were set up for this study, but there are more suboptimal food-related attributes that can be added. For example, the reuse value of suboptimal citrus fruits, damaged packaging when there is packaging, etc., can be used to better understand consumer willingness in relation to price increases or discounts and preferences for different products and attributes; however, expiry dates might make foods more appealing to consumers, but they also represent a restriction for sellers, since they have to waste more unsold food, meaning that this problem has also not been solved here.

This study only explored the consumer dimension, and the results only reflect the current consumer preferences and WTP for suboptimal food. Follow-up research can be aimed at exploring the seller dimension and understanding the opinions of different respondents toward the various attributes of suboptimal food and comparing their differences.

Future research can use a latent class model to test whether there is heterogeneity in respondent preferences for suboptimal foods.

Additionally, this kind of research can apply to the subject of waste to meat and animal products, as these foods are generally the most resource-intensive foods to produce. Therefore, reducing the wastage of these foods represents significant benefits.

## Figures and Tables

**Figure 1 foods-10-00015-f001:**
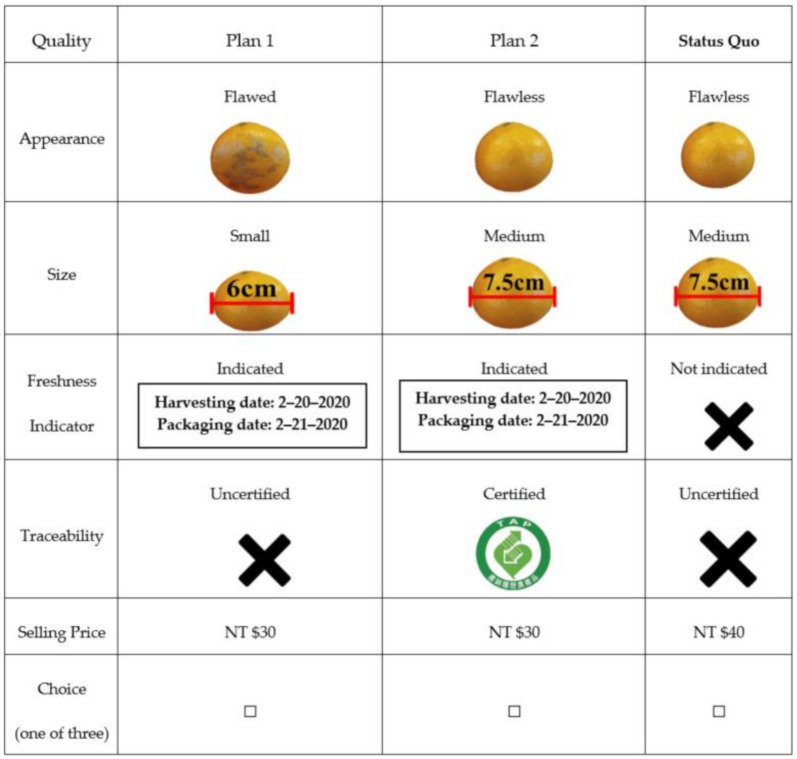
Example questionnaire choice set.

**Table 1 foods-10-00015-t001:** Attributes and levels of suboptimal citrus fruit. NT: New Taiwan dollars.

Attribute	Description of Attribute	Level
Appearance	Compared with optimal citrus fruit, if fruit has spots and a rough or uneven surface then it has a defective appearance.	(1) Complete (2) Defective
Size	According to the standard for the grading and packing of fresh fruit, the size is measured by the circumference length (cm) of the fruit. Small sizes (<23 cm) are usually eliminated.	(1) Large: 25–27 cm (9 cm in diameter) (2) Medium: 23–25 cm (7.5 cm in diameter) (3) Small: 17–23 cm (6 cm in diameter)
Freshness indicator	When citrus fruits are harvested or leave the production area and packaging factory, it is required to mark the time, which can be used by consumers as the basis for measuring the freshness of the product.	(1) Labeled (2) Unlabeled
Traceable agricultural products	Besides avoiding the purchase of citrus fruits from unknown sources, certified citrus fruit has transparent information and a guarantee of origin, safety, and quality, which can be used as a reference for consumers to purchase the product.	(1) Yes (2) No
Price discount	According to a pre-test questionnaire, the discount range was set at $0–15, which was divided into four ranges, including the current situation.	(1) NT $40 (−0) (2) NT $35 (−5) (3) NT $30 (−10) (4) NT $25 (−15)

**Table 2 foods-10-00015-t002:** Conditional logit empirical estimation results. WTP: Willingness to pay.

Attribute Variable	Coefficient	Estimator	*t*-Value	WTP
Keep the status quo (ASC)		1.479	2.41 **	
Appearance (ED)	a1	−1.274	−11.74 ***	NT$18
Small size (SZ2)	a3	−0.244	−1.80 *	NT$36
Freshness indicator (FR)	a4	1.667	14.95 ***	NT$68
Traceability (TAP)	a5	1.376	12.60 ***	NT$63
Price discount (FUND)	β	−0.059	−5.77 ***	

***, **, and, * are significant at 1%, 5%, and 10%, respectively.

**Table 3 foods-10-00015-t003:** Conditional logit (CL) and random parameter logit model (RPL) empirical estimation results.

Attributes and Degrees	Conditional Logit (CL)			Random Parameter Logit (RPL)			
	Coefficient	*t*-Value	Coefficient	*t*-Value	Coefficient Standard Error	*t*-Value	WTP
Appearance (ED)	−1.274	−11.74 ***	−1.853	−9.82 ***	1.046	5.67 ***	NT$19
Large size (SZ1)	−0.053	−0.39	−0.352	−2.03 **	0.222	0.65	NT$36
Small size (SZ2)	−0.244	−1.80 *	−0.646	−3.49 ***	0.385	1.12	NT$33
Freshness indicator (FR)	1.667	14.95 ***	2.142	11.59 ***	1.218	5.86 ***	NT$64
Traceability (TAP)	1.376	12.60 ***	1.724	10.69 ***	0.520	1.63	NT$59
Price discount (FUND)	−0.059	−5.77 ***	−0.089	−6.59 ***			
Choice set count	1455			1455			
Log-likelihood ratio	−1006.131			−951.257			

***, **, and * are significant at 1%, 5%, and 10%, respectively.

**Table 4 foods-10-00015-t004:** Heterogeneity of respondent demographic information in terms of WTP for suboptimal food.

Demographic Information		Sample Size		ASC		ED		FR	
				Average	*t*-Value	Average	*t*-Value	Average	*t*-Value
Gender	Male	248	51%	−6.954	2.173	−21.418	−2.269	23.088	−3.193
	Female	237	49%	−7.762		−20.250		25.214	
Age	20 and below	8	2%	−8.219	2.362 ** (*F*-Value)	−19.807	2.111 (*F*-Value)	24.547	2.480 ** (*F*-Value)
	21–30	96	20%	−8.532		−19.590		26.330	
	31–40	110	23%	−7.355		−20.511		24.108	
	41–50	135	27%	−6.991		−21.043		23.267	
	51–60	125	26%	−6.809		−21.979		23.562	
	61 and over	11	2%	−6.866		−20.681		21.762	
Education	Junior high and below	4	1%	−8.973	1.948 (*F*-Value)	−18.690	1.535 (*F* Value)	24.494	3.196 ** (*F*-Value)
	High school	165	34%	−7.123		−21.375		22.778	
	Tertiary education	235	48%	−7.158		−20.864		24.567	
	Doctorate and above	81	17%	−8.283		−19.828		25.580	
Marital status	Married	279	58%	−6.957	2.455 **	−21.194	−1.561	23.860	−0.924
	Unmarried	206	42%	−7.879		−20.378		24.488	
Average monthly salary	NT$20,000 and below	74	15%	−8.245	2.598 ** (*F*-Value)	−19.587	2.657 ** (*F*-Value)	25.839	2.153 * *(F*-Value)
	NT$20,001–40,000	136	28%	−7.973		−20.149		24.384	
	NT$40,001–60,000	170	35%	−6.679		−21.689		23.751	
	NT$60,001–80,000	80	17%	−7.095		−21.141		23.549	
	NT$80,001 and above	25	5%	−7.429		−19.451		25.366	
Purchase TAP Certified products	Yes	280	58%	−7.291	0.361	−20.829	0.84	23.705	−1.470
	No	205	42%	−7.428		−20.872		24.704	
Scan QR code for information	Will	137	28%	−7.387	−0.127	−20.958	−0.268	23.368	−1.419
	Will not	348	72%	−7.334		−20.804		24.426	

**and * are significant at 5% and 10%, respectively; ASC: keep the status quo; ED: appearance; FR: freshness indicator. TAP: traceability.

## Data Availability

The data presented in this study are available on request from the corresponding author. The data are not publicly available due to the privacy and ethical.

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
