# Peer review of "Assessing Consumer Preferences for Suboptimal Food: Application of a Choice Experiment in Citrus Fruit Retail"

_foods, 2020, doi:10.3390/foods10010015_

Round 1

Reviewer 1 Report

The authors carefully revised the manuscript according to three reviewer's considerations and suggestions. Now, the manuscript is more clear. However, this reviewer strongly suggest English revision prior the publication. Moreover, a perspective on a study focusing on non-buyers of suboptimal food should be added in the conclusion section.

Author Response

Thank you very much for your insightful comments and suggestions. Please refer to attached document.

Reviewer 2 Report

The authors have done a thorough job of integrating most of my feedback. I have just a few further comments.

Table 1: No need to repeat (1) Large, (2) Medium, (3) Small in column 2 and column 3. Delete these from column 2 so that this row is consistent with the others (also move ‘25-27cm…’ to column 3)

Table 4: It would be helpful to have simple percentages within each category as well as N (e.g. 52% male, 48% female, the same for age groups etc.)

Line 454: The point here is to add something to the effect of ‘Governments should relax any regulations on minimum size of fruit, because consumers willl still buy smaller fruits.’

Author Response

(The authors gave the same response as above.)

Reviewer 3 Report

I have read this revision with great interest; the authors answered my questions but I still have concerns about the language. The language definitely needs to be improved. Throughout the paper there are a lot of grammatically incorrect sentences. For example line 41 “ Although the suboptimal food don’t harmful peoples’ health” should be “  Although the suboptimal food doesn’t harm peoples’ health” or the last sentence of the abstract “Based on the result of this study, consumers care freshness indicator the most” should be “Based on the result of this study, consumers care about freshness indicator the most”.

 Also, in the abstract my three previous comments remain unanswered, but I think it is a matter of using correct English language rather than an incorrect study design or study rationale. Food is not suboptimal in traceability and price discounts but these are external attributes that can be used to stimulate the choice of suboptimal food; Also you study “which citrus product attributes construct preferences” instead of “the attributes of suboptimal citrus products”. Suboptimal food is labeled suboptimal because of appearance (which contains size, shape, …), packaging or expiry date. Some of these researches focus on how price discounts or labels can be used to increase purchase intentions/choices of suboptimal food. You research whether providing a discount and a label on traceability will affect choices of food suboptimal in size and flawed appearance. You investigate which attributes people use to construct a preference in that context. Also concerning the last sentence, if freshness is the most important attribute to drive consumer preferences, why is awareness then important? Do you mean awareness about food waste, about health issues,… or about freshness?

Author Response

Thank you very much for your insightful comments and suggestions. Please refer to attached document.

This manuscript is a resubmission of an earlier submission. The following is a list of the peer review reports and author responses from that submission.

Round 1

Reviewer 1 Report

Abstract

It is not clear what you mean with (line 14) a campaign is launched to reduce left overs. Do you mean that throughout the world several campaigns have already been launched in the last couple of years? Why is this important for the topic of the paper (i.e. suboptimal foods). Leftovers is what is left when finishing a meal? Or do you mean with “leftovers” the products that are produced but not sold because they are suboptimal?

Food can be suboptimal in terms of appearance, packaging or expiry date. How can it be suboptimal in traceable agriculture and price discounts? It is not clear in the abstract how the latter are related to suboptimal foods.

How are attributes determined through a choice experiment? Do you mean the importance or use of attributes are determined through the experiment?

I would suggest not to use numbers in the abstract, leave that for that results section (line 25-27). It is not clear what these percentages mean so this makes the abstract not clear.

Line 30-31: not clear. Do you mean that different attributes were determining WTP depending on the socio –economic background?

You end with “therefore, …”. Why is it essential to increase awareness when freshness was the most important determinant? What do you mean with awareness?

Introduction

Numbers on food waste by consumers are provided but the paper is about identifying determinants of buying suboptimal foods so that they would not be wasted by production companies. In this sense, suboptimal foods are not wasted by consumers. Do you have more specific numbers on suboptimal foods and how companies deal with them?

Line 46: Why is it relevant for the paper that Taiwanese people are less aware of waste? Are your results only applicable to Taiwanese people? Or people who are less aware of waste?

Lines 68-82. This paragraph is about which attributes people use for deciding on a product, reasons for waste, reasons for choosing suboptimal products and which communication can increase sales of suboptimal foods. How does the link between attitudes and behavior of suboptimal foods (last sentence) fits within these findings? The last sentence does not seem relevant for the paper.

Line 83-106. Consider restructuring/rewriting this paragraph. It is not clear what you mean to say with this paragraph. The paragraph describes research on sub optimal and ‘normal’ products, WTP, purchase intention for local and organic products, difference between local and organic products, WTP for organic products, WTP for suboptimal foods. Why are organic products relevant for this paper?

The main purpose of the paper is disclosed on line 152-154. It is not clear why looking at the importance of different attributes is relevant? Why those specific attributes like traceability? Are there other attributes that could be relevant? The introduction should clearly state what was found in previous research on determinants of WTP/buying intentions of suboptimal foods and what is missing.

 Methodology

If I understand it correctly you only selected people who already bought suboptimal foods (line 170)? Why? In this way you exclude finding out how to stimulate non-buyers, which is the biggest group of consumers.

You indicate in the introduction that quality is important in driving buying behavior. Did you ask people about their rating of quality for each product?

Figure 1 is not clear, is this what people saw? Furthermore, it is not clear how many products people saw. Did they each see one bundle? What is shown is one bundle? What is condition in figure 1? Are the bundles the same as the sets? What is substitutes and status (line 168)? What is the degree of suboptimality (line 171): did you ask consumers for five products: how important is sub optimality for citrus product 1? For citrus product 2 etc?

Results

Line 232: do you mean that when it comes to suboptimal citrus products, freshness is the most important indicator of their WTP? On line 342 you then indicate that it is about fruit certification attributes?  I cannot assess the validity of the analysis and conclusions since it is not clear to me how the research was performed. The authors should describe more clearly what they did, which questions they asked, what each respondent saw, etc 

Author Response

(The authors gave the same response as above.)

Reviewer 2 Report

Throughout: you can remove the dates for references in this system - just the numbers in square parentheses is sufficient.

Abstract

Introduction

Line 50: I would add the words ‘being near the expiry date’

Materials and Methods

Line 160: Use past tense here (‘This study explored’)

Table 1: The sentence in row 2 is superfluous: ‘This study is intended to explore consumers’ preferences for suboptimal citrus’. Also, no need to repeat the levels in column 2 and 3 - I suggest moving the measurements into column 3

Table 1: Please explain the abbreviation ‘NT’ in the last row.

Line 168: What is meant by ‘one status’?

Line 179: These characteristics would be better described as ‘demographic information’ rather than ‘socioeconomic background’, which tends to refer only to income, wealth and employment

Results and Discussion

Line 215-217: No capitalisation needed for ‘Judgemental Sampling’ and ‘Suboptimal Food’

Line 218: It may be clearer to display this demographic information in a table.

Line 227: It would be helpful to have an explanation of what is meant by ‘mass merchandisers’

Line 248: Extra word ‘certification’ here - also the 2.27% is repeated

Table 2: It is not clear what the status quo represents here - this is the first time this is mentioned

Line 317: I would think that the reason for preferring foods with a normal appearance is inferences about taste, texture, and safety rather than simply greater exposure in retail environments

Conclusions

The first paragraph here is copied from above! Re-write this first paragraph so that it is not exactly the same

Line 362: Since you found that size was the least important factor, it seems to follow that these restrictions on size should be relaxed (since consumers will still buy smaller fruits)

Line 393: You can also recommend to do similar studies on different types of food. In particular, reducing waste of meat and animal products would be very beneficial, since these foods are generally the most resource-intensive to produce.

Line 397: Yes - in particular, expiry dates might make foods more appealing to consumers, but represent a restriction for sellers, since they have to waste more unsold food

Author Response

(The authors gave the same response as above.)

Reviewer 3 Report

Revision Manuscript n. 925450

Journal: Foods

Title: Assessing Consumer Preferences for Suboptimal Food: A Choice Experiment Application

The manuscript entitled “Assessing Consumer Preferences for Suboptimal Food: A Choice Experiment Application” describes the consumer preferences towards suboptimal citrus fruit in Taiwan. In addition, conditional logistic (CL) regression and random parameter logit model (RPLM) were employed to examine various properties of the marginal willingness to pay (WTP). The manuscript is very interesting and deals with food security issues. In my opinion, the experimental plan is well defined and the results clear presented. I suggest minor changes.

L2-3 Replace with “Assessing Consumer Preferences for Suboptimal Food: A Choice Experiment Application During Citrus Fruit Retail”

L36-157 Although the Introduction section includes relevant works on suboptimal foods and willingness to pay I suggest to short this section. Actually the section includes ca. 1900 words. Please short to ca. 1700 words. Thanks

L43-44 Please revise this sentence. It is not clear

L152-153 Please include the following sentence:

In addition, citrus fruit appearance is limited by fungal decay [X] and peel defects [Y] during store retailing, affecting the willingness to pay.

  1. X) Pinto, L.; Cefola, M.; Bonifacio, M.A.; Cometa, S.; Bocchino, C.; Pace, B.; De Giglio, E.; Palumbo, M.; Sada, A.; Logrieco, A.F.; Baruzzi, F. Effect of red thyme oil (Thymus vulgaris) vapours on fungal decay, quality parameters and shelf-life of oranges during cold storage. Food Chem. 2021, 336, 127590. doi.org/10.1016/j.foodchem.2020.127590
  2. Y) Cronjé, P.J.; Zacarías, L.; Alférez, F. Susceptibility to postharvest peel pitting in Citrus fruits as related to albedo thickness, water loss and phospholipase activity. Postharvest Biol. Technol. 2017, 123, 77-82. doi.org/10.1016/j.postharvbio.2016.08.012

L187-188 Please rewrite the sentence. It is not clear

L196 kth. Please check

L197 are the parameter…

L201 Please check ith and kth. Are they parameters? Please explain

L206-207 Please rewrite the sentence. It is not clear

L217-218 Please specify how the invalid questionnaires were selected.

L220-221 This is a result. It should be reported in the Results and Discussion section

L269 Please explain in the text “status quo” meaning.

L275-277 Please rewrite. It is not clear

L279-283 Please rewrite. It is not clear. I suggest using short sentences

L291-292 Please rewrite. It is not clear.

L300 Replace semi colon with comma

L300-302 Please explain better these results. They are not clear

Table 3 Please clear separate the parameters belonging to Conditional Logit or Random Parameter Logit. The Table is not clear in this form. Thanks

Table 4 Please indicate the meaning of ASC, ED and FR parameters. Thanks

L386-388 Please delete these sentences.

Author Response

(The authors gave the same response as above.)
